# An Evaluation of Laminarin Additive in the Diets of Juvenile Largemouth Bass (*Micropterus salmoides*): Growth, Antioxidant Capacity, Immune Response and Intestinal Microbiota

**DOI:** 10.3390/ani13030459

**Published:** 2023-01-28

**Authors:** Youjun Wu, Yan Cheng, Shichao Qian, Wei Zhang, Mengmeng Huang, Shun Yang, Hui Fei

**Affiliations:** 1College of Life Sciences and Medicine, Zhejiang Sci-Tech University, Hangzhou 310018, China; 2Huzhou Baijiayu Biotech Co., Ltd., Huzhou 313000, China; 3Post-Doctoral Innovation Practice Base of South China Agricultural University and South China Normal University, Guangzhou 511466, China

**Keywords:** *Micropterus salmoides*, laminarin, antioxidant capacity, intestinal microbiota, immune response

## Abstract

**Simple Summary:**

Due to consumer demand for safe aquatic foods and environmental concerns in aquaculture, aquaculture production can benefit from investigating the use of immunostimulant b-glucans as feed additives. In this study, we selected laminarin as a feed additive for juvenile largemouth bass (*Micropterus salmoides*), and investigated the appropriate dose of supplemented laminarin in the diet based on the analysis of the growth performance, antioxidant capacity, immune response and intestinal microbiota of fish. The results indicated that supplemented laminarin in diet at a low level is suggested as a promising immunopotentiator without negative effects on growth performance for juvenile largemouth bass.

**Abstract:**

A 28 day feeding trial was conducted to investigate the growth performance, immune response and intestinal microbiota of laminarin (LAM) supplemented diets in juvenile largemouth bass (*Micropterus salmoides*). Four hundred and eighty fish (initial average weight: 0.72 ± 0.04 g) were randomly divided into four groups (40 fish per tank with three replicates in each group) Four diets were prepared with LAM supplementation at the doses of 0 (control), 5 g Kg^−1^ (LL), 10 g Kg^−1^ (ML) and 15 g Kg^−1^ (HL), respectively. No significant difference in the specific growth rate (SGR) and hepatosomatic index (HSI) was observed in fish among the four groups, or in the lipid and ash content of fish flesh. In addition, fish in the LL group exhibited much higher antioxidant capacity (*p* < 0.05), while the diets with the inclusion of 5 and 10 g Kg^−1^ LAM remarkably decreased the antioxidant capacity of fish (*p* > 0.05). Dietary LAM at the dose of 5 g Kg^−1^ inhibited the transcription of interleukin-1β (*il-1β*) and tumor necrosis factor-α (*tnf-α*), while promoting the expression of transforming growth factor-β (*tgf-β*) in fish intestine. Moreover, the beneficial intestinal bacteria *Bacteroide*, *Comamonas* and *Mycoplasma* abundance significantly increased in fish from the LL group, while the content of opportunistic pathogens *Plesiomonas*, *Aeromonas* and *Brevinema* in fish of the HL group was substantially higher than the control group. Overall, the appropriate dose of supplemented LAM in the diet was 5 g Kg^−1^, while an excessive supplementation of LAM in the diet led to microbial community instability in largemouth bass.

## 1. Introduction

In the last decades, largemouth bass (*Micropterus salmoides*) has been an important economic fish and is widely accepted by consumers all over the world [1,2,3]. The frequent outbreaks of diseases caused by various pathogens have become a limiting factor for the development of largemouth bass farming [3]. The application of antibiotics effectively decreases the outbreaks of diseases, while it presents harmful effects, such as the development of resistant bacteria and the accumulation in the natural environment [4]. With the demand of consumers for safe aquatic food, as well as the requirement of environmental security, there is a prerequisite to develop safe and efficient dietary additives that can promote the physiology and health of the farmed aquatic animals.

Laminarin (LAM) is a β-glucan extracted from brown algae, and is composed of β-1, 3-glucan with β-1, 6-linkages [5]. Lines of evidence have reported that specific physicochemical properties play a vital role in determining the magnitude of β-glucan binding to macrophage receptor(s) and how it modulates the immune responses [6]. Moreover, published reports have demonstrated that LAM features antioxidant, immunopotentiator, antitumor and antivirus properties [7]. In addition, a large number of research works revealed that LAM displayed immune-modulatory effects in fish because of its binding capacity to different receptors on leukocytes, leading to the stimulation of immune responses including bactericidal activity, cytokine productivity, and survival fit ability at cellular levels [7,8,9,10,11]. For example, dietary LAM significantly increased the alkaline phosphatase activity, as well as enhanced the superoxide dismutase (SOD) activity of the Pearl gentian grouper [7]. Similarly, the supplementation of LAM in the diet significantly improved the SGR, as well as the lysozyme (LZM), catalase and superoxide dismutase activities of the grouper (*Epinephelus coioides*) [9]. More recently, Jiang et al. demonstrated that the expression levels of Toll-like receptor 5 and insulin-like growth factor 2 were remarkably promoted in Channel Catfish (*Ictalurus punctatus*) fed laminarin at a dose of 4 g kg^−1^ [11]. The above findings suggested that LAM could act as a beneficial supplement in the fish diet. However, little information is currently available about the impact of dietary LAM on the physiology and immune response of juvenile largemouth bass.

Furthermore, growing evidence revealed that dietary β-glucan could affect the intestinal flora of fish [12,13,14,15,16]. Meanwhile, the diversity and richness of intestinal microbiota affect a wide range of host physiological states including growth performance and immune response of the aquatic host [17,18,19]. For instance, Jung et al. stated that dietary β-glucan increased the diversity of the intestinal microbial community, which helps common carp (*Cyprinus carpio*) to prevent pathogenic microbes invasion [13]. Similarly, metagenomic analysis revealed that dietary β-glucan water (0.1 mg L^−1^) sharply increased the Chao richness value (*p* < 0.05), with a larger content of the *Vibrionaceae* family in Nile tilapia (*Oreochromis niloticus*) [15]. By contrast, a reduction of the intestinal microbiota richness was observed in common carp fed a β-glucan inclusion diet [12]. Therefore, the effect of dietary LAM on juvenile largemouth bass may also be associated with the intestinal microbiota variation.

In this study, four experimental diets were prepared with LAM supplementation levels of 0 (control), 5 g Kg^−1^ (LL), 10 g Kg^−1^ (ML) and 15 g Kg^−1^ (HL). A 28 day feeding trial was conducted to evaluate the effect of LAM on growth performance, antioxidant capacity and immune response, and explore how LAM influences the intestinal microbiota of juvenile largemouth bass.

## 2. Materials and Methods

### 2.1. Experimental Diets

Laminarin was purchased from Xiya Reagent (Xiya Reagent Co., Ltd., Linyi, China) with a purity of at least 99.5%, and laminarin was extracted from *Laminaria digitateusing* using the warm-water extraction method. Four diets were prepared with the supplementation of LAM: 0 (control), 5 g Kg^−1^ (LL group), 10 g Kg^−1^ (ML group) and 15 g Kg^−1^ (HL group). According to the formulation (Table 1), all ingredients were crushed through a 200 μm mesh and then blended with the fish oil and water through a mixing device [3]. Subsequently, pellets with a size of 1.0–2.0 mm were produced by using the pelletizer (KCHL-10, Kcth Group, Beijing, China) [19]. Then, the particle feed was dried in the air thermal dryer (50 ℃) (Longhe Machinery, Chaozhou, China) for further use.

### 2.2. Experimental Design

Juvenile largemouth bass (initial average weight: 0.72 ± 0.04 g) were obtained from Huzhou Baijiayu Biotech Co., Ltd. (Huzhou, China). The fish were cultured in circulating resin tanks with a constant water flow (100 L h^−1^) during the experimental period. Juvenile fish were fed with a commercial diet (Tianma Co., Ltd., Fuzhou, China) for a week before being fed experimental diets. A total of 480 juvenile fish were randomly assigned to four groups (40 fish per tank with 3 replicates in each group). They were fed regularly thrice daily (8:00 a.m., 14:00 p.m. and 20:00 p.m.) until apparent satiation. The feeding trial lasted 28 days, with the water temperature at 27.0 ± 1.0 ℃, pH = 7.2–7.5 and dissolved oxygen < 5.0 mg L^−1^.

### 2.3. Sampling

Before sampling, all juvenile largemouth bass were fasted for 24 h, and then anesthetized with tricaine methanesulfonate (MS-222) at a dose of 55 mg/L. Subsequently, all fish were sampled and sacrificed to measure the individual physiology. Twenty-four fish per replicate were dissected under sterile conditions to pull out the intestine, and then the mid intestine was cut into small pieces and washed with phosphate-buffered saline (PBS) (pH 7.5) to harvest the intestinal tissues and contents (four samples (each sample contains 6 fish) in each tank). Then, both of them were immediately stored at −80 °C in TRIzol reagent (Tiangen, Beijing, China) for RNA extraction. Nine fish per replicate were sampled and frozen by liquid nitrogen, and was stored in a −80 °C refrigerator for the analysis of flesh composition and antioxidant capacity, respectively.

### 2.4. Flesh Composition and Antioxidant Capacity Analysis 

The AOAC method (AOAC, 2000) was used to analyze the proximate composition of flesh following our previous report [3]. The antioxidant indices of intestine, including superoxide dismutase (SOD), catalase (CAT), glutathione (GSH) and total antioxidant capacity (T-AOC), were measured by using commercial kits obtained from Jian Cheng Bioengineering Institute, (Nanjing, China) [20]. 

### 2.5. Relative Gene Expression Analysis

The relative expression of genes related to immune response in intestine was measured as described in our previous report [19]. Briefly, the primer sequences of tgf-β, tnf-α, il-1β and β-actin were designed as listed in Table 2. Total RNA was extracted from the intestinal tissues using Trizol reagent (Tiangen, Beijing, China). The final RNA was eluted in an appropriate amount of 0.1% diethyl pyrocarbonate (DEPC) treated water (Sigma-Aldrich, St. Louis, MO, USA). The RNA amount was determined using a Nanodrop 2000. Then, the cDNA was synthesized with the reverse transcription [19]. Subsequently, RT-qPCR was operated with following steps: 95 °C for 30 s; 40 cycles of 95 °C for 5 s, 60 °C for 30 s, and 72 °C for 30 s. Expression of the selected genes was normalized to β-actin (internal reference) and reported as 2^−ΔΔCt^ [21]. 

### 2.6. Sequencing of Intestinal Microorganisms

The intestinal contents were prepared for 16S rRNA sequence [19]. Briefly, the primers 338F 5′-ACTCCTACGGGAGGCAGCAG-3′ and 806R 5′-GGACTACHVGGTW TTAAT-3′ were used to amplify the V3-V4 region of the 16S ribosomal RNA gene. After purifying the amplicon DNA, the SMRTbell libraries were established by blunt-end ligation, which were then sequenced by Biomarker Technologies (Beijing, China). Finally, the intestinal microbiota analysis, including Principal component analysis (PCA), Venn diagram, Microbial community bar plots (MCBP) and Linear discriminant analysis Effect Size (LEfSe) were performed using the BMKCloud software (www.biocloud.net) (accessed on 8 November 2022).

### 2.7. Statistics Analysis

The data were shown as the mean ± standard deviations. Statistical analysis was performed by one-way ANOVA and Duncan’s multiple range test (DMRT), using the SPSS software (Version 20.0; SPSS, Inc). Differences with *p* < 0.05 were regarded to be statistically significant.

## 3. Results

### 3.1. Growth Performance

At the end of the feeding trial, the SGR of juvenile largemouth bass in the LL group was slightly higher than that of control, while dietary LAM inclusion at a dose of 15 g Kg^−1^ (HL group) remarkably increased the feed conversation rate (FCR) (*p* < 0.05), as well as marginally decreasing the SGR (*p* > 0.05). Moreover, juvenile fish in the LL group exhibited a considerably higher condition factor (CF) than that of fish in the ML and HL groups (*p* < 0.05). Meanwhile, no remarkable difference in HSI was observed among the four groups (*p* > 0.05) (Table 3).

### 3.2. Flesh Composition

In this study, no significance of moisture and crude lipid content in fish flesh among four groups was observed (*p* > 0.05). Nevertheless, the minimum of flesh crude protein content was observed in the HL group (*p* < 0.05) (Table 4).

### 3.3. Antioxidant-related Enzyme Activities

Compared with the control group, juvenile largemouth bass in the LL group exhibited considerably higher SOD, GSH and T-AOC activity, while fish in the ML and HL groups displayed significantly lower SOD and CAT activity (*p <* 0.05) (Figure 1).

### 3.4. Relative Expression Levels of Immune Response Related Genes

The expression of *il-1β* and *tnf-α* was dramatically down-regulated in fish of the LL group, while it sharply increased in fish of the ML and HL groups (*p* < 0.05). Meanwhile, dietary LAM inclusion substantially increased the expression of *tgf-β* in fish of the LL group (*p* < 0.05) (Figure 2).

### 3.5. Intestinal Microbiota Changes

The 16 sRNA sequence analysis showed that 1,320,788 raw reads and 1,316,678 clean reads were obtained from 16 samples, with an average of 82,292 clean reads for each sample (Appendix A). The microbiota diversity was calculated from the OTUs (Table 5), which revealed that these four indexes were significantly affected in LAM supplemented groups, where fish in the HL group exhibited the minimum values (*p <* 0.05). The heat map showed that the dominant genera were altered in LAM inclusion groups and the intra group differences are acceptable (Figure 3). Additionally, the PCA showed that the intestinal microbiota of fish in four experimental groups was grouped separately (Figure 4A).

Moreover, the Venn diagram showed that fish in the control group had the largest number of OTU, while fish in the LL group exhibited the minimum OTU (Figure 4B). MCBP displayed that Proteobacteria was the dominant phyla in the control and LL groups, followed by Firmicutes and Bacteroidetes at phylum level (Figure 5A). The Firmicutes/Bacteroidetes ratio in the control, LL, ML and HL groups was 1.40, 4.00, 0.88 and 19.05, respectively (Appendix A). Moreover, unclassified_*Bacteria* was the major taxon, followed by unclassified_*Cyanobacteriales* and *Bacteroides* in the control group. *Mycoplasma* was the main taxon in the LL group, followed by *Comamonas* and *Plesiomonas*. *Plesiomonas* was the dominate taxon in the ML group, followed by unclassified_*Cyanobacteriales* and *Cyanobium.* In the HL group, *Aeromonas* was the primary taxon, followed by *Plesiomonas* and *Mycoplasma* at genus level (Figure 5B). The LEfSe analysis showed that the *Bacteroide*, *Comamonas* and *Mycoplasma* content in the LL group, and the *Aeromonas*, *Brevinama* and *Plesiomona* content in the HL groups, was remarkably higher than that of the control at genus level (*p* < 0.05) (Figure 5C–E).

## 4. Discussion

LAM supplementation has been shown to improve growth in a variety of animals, including weaned piglets [22,23,24], grouper (*E. coioides*) [9], Channel Catfish (*I. punctatus*) [11], and Pearl gentian grouper (*Epinephelus lanceolatus* ♀ × *Epinephelus fuscoguttatuss* ♂) [6]. In aquatic animals, previous study has demonstrated that dietary LAM inclusion at 5 and 10 g kg^−1^ dramatically increased the WGR of grouper (mean weight 90 ± 2.6 g) compared to that of fish in the 15 g kg^−1^ inclusion group and control group for 48 days [9]. Similar results have been determined in studies of Channel Catfish (average weight is 1.3 ± 0.3 g), with a significant increase of WGR and SGR in fish fed LAM at 4 and 8 g kg^−1^ for 45 days [11]. In our study, no significant differences of WGR and SGR were observed in juvenile largemouth bass fed with LAM (*p* > 0.05), while dietary LAM inclusion at 15 g kg^−1^ considerably increased the FCR (*p* < 0.05) for 28 days. These discrepancies may be attributable to the fish species and developmental stages difference, as well as the short period of the feeding trial in this study, so the effect of LAM on juvenile largemouth bass in a longer feeding trail (8 weeks or more) should be investigated in further research. Meanwhile, the results also indicated that a high level of LAM additive in the diet may negatively affect fish growth and other physiological functions. In addition, juvenile fish in the LL group exhibited significantly higher CF than that of fish in the ML and HL groups (*p* < 0.05), which might be due to the higher concentration of crude protein and lipids in the diet of the LL group.

β-glucan has been previously assessed as an immunopotentiating agent for enhancing fish immunity to stress and disease [24,25,26,27,28]. Antioxidant capacity is the fundamental cytoprotective mechanism against oxidative stress in fish tissues [29,30]. Previous reports have demonstrated that a higher level of activities of SOD and CAT was determined in common carp after intraperitoneal injection with β-glucan for 15 days [26]. As an oral additive, fucoidan (10 g kg^−1^) and *Halymenia 9ilatate*-derived polysaccharide (1.0 and 2.0  g kg^−1^) supplemented in diet enhanced the antioxidant capacity of *O. niloticus*, respectively [31,32]. Similarly, dietary chitosan (5.0 g kg^−1^ feed) also significantly increased the antioxidant enzyme activity in loach fish (*Misgurnus anguillicadatus*) [33]. Consistently, in our work, dietary LAM at a dose of 5.0 g kg^−1^ remarkably enhanced the T-AOC, SOD and GSH activity, while high dose supplementation (10 and 15 g kg^−1^) significantly decreased the CAT and SOD activity. This tendency of the low concentration to promote antioxidant capacity and high concentration inhibition antioxidant capacity was similar to that of the experiment in red swamp crayfish [34]. The reason that higher levels of LAM supplementation in the diet decreased the CAT and SOD activity might be that the feed composition greatly changed, which should be investigated in further research.

Receptor-bound β-glucan may mediate the production of inflammatory cytokines (interleukins, interferons and tumor necrosis factor). These signaling proteins are believed to aggravate the phagocytic activity of immune cells through oxidative burst and natural cytotoxic liquidation [6]. *tnf-α* is a key pro-inflammatory cytokine, which acts as an important mediator in the regulation of inflammatory response, and induces the gene expression of some pro-inflammatory factors, such as *il-1β* [35]. A published report demonstrated that lower expression patterns of *il-1β* and *tnf-α* were found in juvenile carp fed with β-glucan supplements (6 mg kg^−1^ body weight) for 14 days [36]. Similar depressed expression patterns of *il-1β* and *tnf-α* were also determined in turbot (*Scophthalmus maximus*) fed inclusion with β-glucan for 24 days [13]. Consistently, the expression of *tnf-α* and *il-1β* were significantly decreased in largemouth bass fed a diet containing 5 g Kg^−1^ LAM for 28 days in the present study. Nevertheless, Yin et al. informed that dietary LAM at a low dose (0.5%) for 48 days sharply reduced the mRNA level of *il-1β* in grouper [8]. Collectively, the above finds suggested that the effect of LAM on fish immune related gene expression might rely on the fish species and administration period. Moreover, dietary LAM at a low dose (5 g Kg^−1^) remarkably increased the expression of *tgf-β* in this work. It was well demonstrated that *tgf-β* could depress the production of pro-inflammatory cytokine, and further inhibit the inflammatory response in teleost [37]. This opposite change trend of *il-1β* and *tgf-β* expression was similar to the result in some other fish species [38,39]. Overall, the down-regulated expression of *il-1β* and *tnf-α*, up-regulated expression of *tgf-β*, as well as the elevated antioxidant enzyme activity, confirmed the contribution of dietary LAM inclusion at the concentration of 5 g Kg^−1^ in enhancing the immune response and antioxidant capacity of juvenile largemouth bass.

Notably, dietary LAM inclusion can modify intestinal microbiota in the present study. Our microbiota analysis displayed that Proteobacteria and Firmicutes were the two dominant taxon in the control and LL groups at phylum level, which is consistent with published reports on juvenile largemouth bass [20,40,41]. However, with the increase of LAM addition, the Proteobacteria content remarkably increased in ML and LL groups. Published reports have confirmed that an increase in Proteobacteria is often considered an important symbol of intestinal microbiota instability, which may give rise to nutritional and metabolic disorders of the host [41,42]. Moreover, the Firmicutes/Bacteroidetes ratio was 1.40, 4.00, 0.88 and 19.05 in the Con, LL, ML and HL groups, respectively. It was reported that the Firmicutes/Bacteroidetes ration reflected the ability of nutrient transportation and utilization [43,44], while in the HL group most of the identified species of Proteobacteria were *Plesiomonas, Aeromonas* and *Brevinema*, which usually exist in intestines of aquatic animals, and are recognized as potential pathogens of fishes [45,46,47]. Moreover, our data also revealed that the *Mycoplasma*, *Bacteroide* and *Comamonas* abundance in LL groups was remarkably higher than that of the control. Growing evidence showed that *Mycoplasma* was the major species existing in healthy largemouth bass intestine [3,40,48,49,50,51], which might play a certain role in the growth and reproduction of fish [52,53]. *Bacteroides* can generate many organic acids [54,55], which have been evidenced successfully in alleviating intestinal inflammation in fish [56]. *Comamonas* is extensively distributed in soil and contributes to organic biodegradation by reducing Fe^3+^/HS, which is considered a beneficial intestinal bacteria [57] and can be employed as a probiotic additive [58]. Overall, these results suggested that dietary LAM (5 g Kg^−1^) can increase the beneficial bacteria abundance in the intestine and, further, may positively affect the physiological performance, while an excessive addition of LAM led to an increase of pathogenic bacteria in the juvenile largemouth bass intestine.

## 5. Conclusions

The present complementary analysis of growth performance, immune response and intestinal microbiota in LAM supplemented diets in juvenile largemouth bass indicated that the supplementation of the LAM at the dose of 5 g Kg^−1^ is suggested as a promising immunopotentiator without negative effects on the growth performance for juvenile largemouth bass.

## Figures and Tables

**Figure 1 animals-13-00459-f001:**
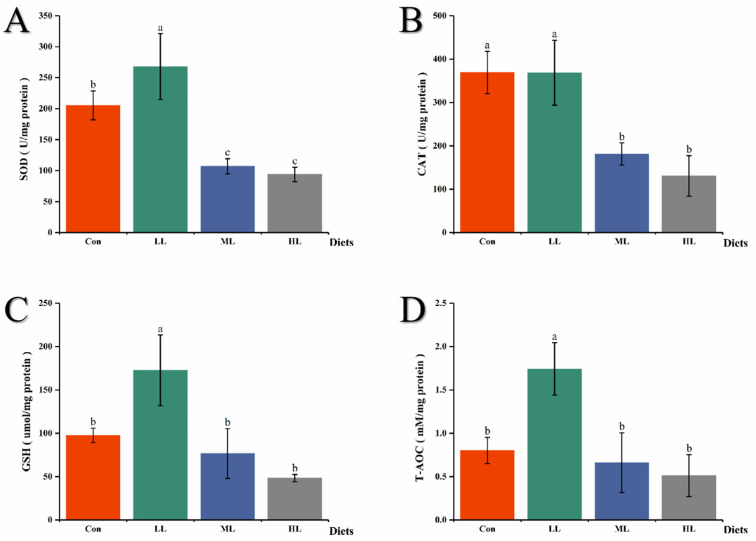
Antioxidant enzyme activities of the intestine in juvenile largemouth bass. (**A**) Superoxide dismutase; (**B**) Catalase; (**C**) Glutathione; (**D**) Total antioxidant capacity. Con means control diet. LL, ML, and HL mean 5 g Kg^−1^, 10 g Kg^−1^ and 15 g Kg^−1^ laminarin was supplemented into diets, respectively. Results are expressed as mean ± SEM (*n* = 3). Means in the same row with different letters were significantly different among groups (*p* < 0.05).

**Figure 2 animals-13-00459-f002:**
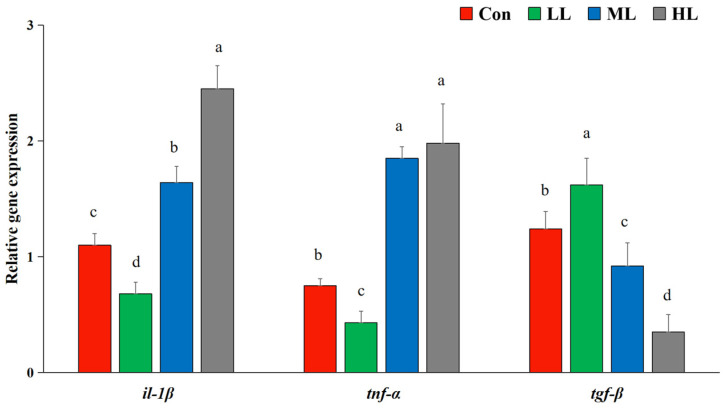
Effects of dietary laminarin inclusion on transcriptional expression of juvenile largemouth bass intestine. Results are expressed as mean ± SEM (*n* = 3). Con means control diet. LL, ML, and HL mean 5 g Kg^−1^, 10 g Kg^−1^ and 15 g Kg^−1^ laminarin was supplemented into diets, respectively. Results are expressed as mean ± SEM (*n* = 3). Means in the same row with different letters were significantly different among groups (*p* < 0.05).

**Figure 3 animals-13-00459-f003:**
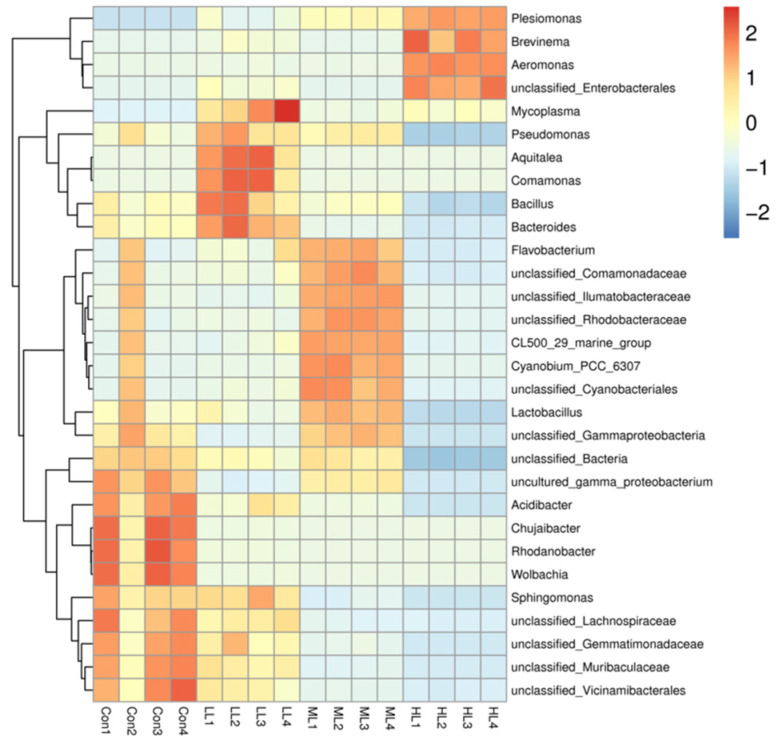
The heat map of intestinal microbiota of largemouth bass fed with four experimental diets. Con means control diet. LL, ML, and HL mean 5 g Kg^−1^, 10 g Kg^−1^ and 15 g Kg^−1^ laminarin was supplemented into diets, respectively.

**Figure 4 animals-13-00459-f004:**
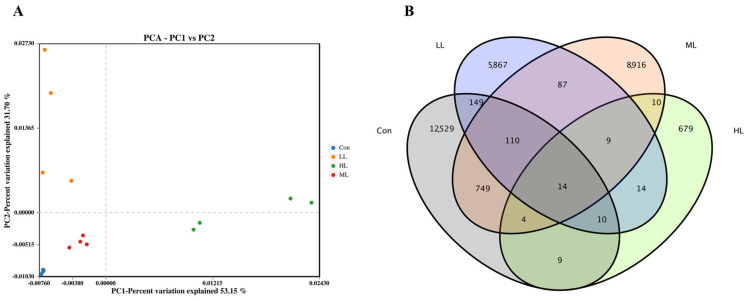
The principal component analysis and Venn diagram of intestinal microbiota of largemouth bass. (**A**) The principal component analysis. (**B**) The Venn diagram. Con means control diet. LL, ML, and HL mean 5 g Kg^−1^, 10 g Kg^−1^ and 15 g Kg^−1^ laminarin was supplemented into diets, respectively.

**Figure 5 animals-13-00459-f005:**
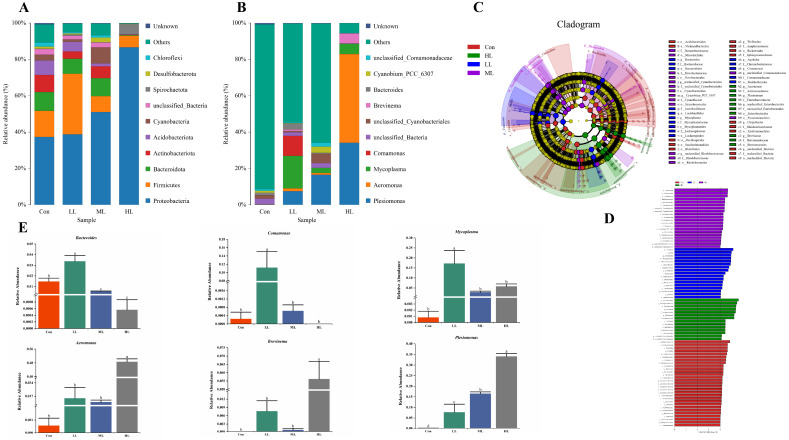
Intestinal microbiota changes between control and laminarin supplemented groups of juvenile largemouth bass. (**A**) The bar plots at phylum level. (**B**) The bar plots at genus level. (**C**) The Lefse analysis of differential flora between control and laminarin Scheme 4. 0. (**D**) The LDA scores of different microbes between control and laminarin supplemented groups. (**E**) The relative abundance of microbes with significant difference. Values in rows with different superscript letters have significant differences (*p* < 0.05). Con means control diet. LL, ML, and HL mean 5 g Kg^−1^, 10 g Kg^−1^ and 15 g Kg^−1^ laminarin was supplemented into diets, respectively.

**Table 1 animals-13-00459-t001:** Formulation and proximate composition of the experimental diets *.

Ingredients (g kg^−1^)	Diets			
	Con	LL	ML	HL
Fishmeal	550	550	550	550
Shrimp meal	100	100	100	100
Laminarin	0	5	10	15
Corn protein	130	130	130	130
Soybean meal	60	60	60	60
Soluble starch	40	40	40	40
Fish oil	80	80	80	80
Ca(H_2_PO_4_)_2_	10	10	10	10
Mineral premix ^1^	10	10	10	10
Vitamin premix ^2^	8	8	8	8
Cr_2_O_3_	4	4	4	4
Choline chloride	3	3	3	3
Threonine	3	3	3	3
Methionine	2	2	2	2
Proximate composition (%)				
Dry matter	94.11	94.17	94.03	94.18
Crude protein	49.77	49.43	49.21	49.09
Crude lipid	9.53	9.50	9.44	9.39
Ash	10.78	10.62	10.72	10.69

* Con means control diet. LL, ML, and HL means 5 g Kg^−1^, 10 g Kg^−1^ and 15 g Kg^−1^ laminarin was supplemented into diets, respectively. ^1^ One kilogram of vitamin premix provided: zeolite, 638 mg; FeSO_4_·H_2_O, 300 mg; ZnSO_4_·H_2_O, 200 mg; MnSO_4_·H_2_O, 100 mg; NaCl, 100 mg; KIO_3_ (10%), 80 mg; Na_2_SeO_3_ (10% Se), 67 mg; CuSO_4_·5H_2_O, 10 mg; CoCl_2_·6H_2_O, 5 mg. ^2^ One kilogram of vitamin premix provided: vitamin C, 400 mg; vitamin E, 200 mg; inositol, 200 mg; niacinamide, 100 mg; calcium pantothenate, 40 mg; vitamin A, 20 mg; vitamin B6, 15 mg; vitamin B1, 12 mg; vitamin B2, 10 mg; folic acid, 10 mg; vitamin K3, 10 mg; vitamin D3, 10 mg; vitamin B12 (1%), 8 mg; biotin (2%), 2 mg.

**Table 2 animals-13-00459-t002:** Real-time PCR primers sequences *.

Gene	Sequence (5′-3′)	Tm (℃)	Product Size (bp)	Accession Number
*il-1β*	F: CGTGACTGACAGCAAAAAGAGG	60	166	XM_038733429.1
	R: GATGCCCAGAGCCACAGTTC	61		
*tnf-α*	F: CTTCGTCTACAGCCAGGCATCG	63	162	XM_038710731.1
	R: TTTGGCCACACCGACCTCACC	65		
*tgf-β*	F: GCTCAAAGAGAGCGAGGATG	58	118	XM_038693206.1
	R: TCCTCTACCATTCGCAATCC	57		
*β-actin*	F: TGGAAGGGACCTCACAGACTAC	61	231	MH018565
	R: GGGCAACGGAACCTCTCAT	60		

* *il-1β*, interleukin-1β; *tnf-α*, tumor necrosis factor-α; *tgf-β*, transforming growth factor-β.

**Table 3 animals-13-00459-t003:** Growth performance of juvenile largemouth bass fed the experimental diets for 28 days *.

Item	Con	LL	ML	HL
IBW(g)	0.72 ± 0.04	0.73 ± 0.02	0.72 ± 0.04	0.71 ± 0.04
FBW(g)	2.88 ± 0.21	3.00 ± 0.15	2.84 ± 0.30	2.75 ± 0.30
WGR(%)	297.38 ± 29.58	311.81 ± 19.85	296.37 ± 42.18	285.15 ± 42.30
SGR(%/day)	4.92 ± 0.56	5.05 ± 0.17	4.90 ± 0.37	4.80 ± 0.40
Total feed intake (g)	90.72 ± 2.12	92.62 ± 1.34	88. 19 ± 1.71	90.58 ± 1.86
FCR	1.05 ± 0.02 ^b^	1.02 ± 0.04 ^b^	1.04 ± 0.03 ^b^	1.11 ± 0.02 ^a^
CF(g/cm³)	1.04 ± 0.07 ^ab^	1.07 ± 0.07 ^a^	0.98 ± 0.06 ^b^	1.00 ± 0.05 ^b^
HSI(%)	1.76 ± 0.28	1.64 ± 0.04	1.67 ± 0.19	1.69 ± 0.14

* Data are shown as mean ± standard error (SEM; *n* = 3). Values in the same row with different superscripts are significantly different (*p* < 0.05). IBW means the initial body weight; FBW means the final body weight; WGR means the weight gain rate; SGR means the specific growth rate; FCR means the feed conversation rate; HSI means the hepatosomatic index; CF means the condition factor. Con means control diet. LL, ML, and HL mean 5 g Kg^−1^, 10 g Kg^−1^ and 15 g Kg^−1^ laminarin was supplemented into diets, respectively.

**Table 4 animals-13-00459-t004:** Effects of dietary laminarin inclusion on flesh composition of juvenile largemouth bass fed the experimental diets for 28 days *.

Item	Con	LL	ML	HL
Moisture (%)	73.54 ± 0.36	73.97 ± 0.17	73.86 ± 0.22	73.50 ± 0.31
Crude protein (%)	15.74 ± 0.45 ^a^	16.41 ± 0.38 ^a^	15.72 ± 0.33 ^ab^	15.42 ± 0.29 ^b^
Crude lipid (%)	3.76 ± 0.14	3.84 ± 0.07	3.75 ± 0.07	3.72 ± 0.09

* Data are shown as mean ± standard error (SEM; *n* = 3). Values in the same row with different superscripts are significantly different (*p* < 0.05). Con means control diet. LL, ML, and HL mean 5 g Kg^−1^, 10 g Kg^−1^ and 15 g Kg^−1^ laminarin was supplemented into diets, respectively.

**Table 5 animals-13-00459-t005:** Alpha diversity of intestinal microbiota of juvenile largemouth bass fed the experimental diets for 28 days *.

Item	Con	LL	ML	HL
ACE	3931.84 ± 735.78 ^a^	1722.66 ± 130.31 ^c^	2976.99 ± 137.85 ^b^	221.94 ± 43.89 ^d^
Shannon	10.74 ± 0.16 ^a^	8.00 ± 0.27 ^c^	9.03 ± 0.04 ^b^	2.91 ± 0.14 ^d^
Simpson	1.00 ± 0.00 ^a^	0.97 ± 0.02 ^b^	0.99 ± 0.00 ^a^	0.78 ± 0.01 ^c^
Chao1	3927.94 ± 732.99 ^a^	1720.83 ± 130.43 ^c^	2966.06 ± 138.33 ^b^	220.46 ± 44.03 ^d^

* Con means control diet. LL, ML, and HL mean 5 g Kg^−1^, 10 g Kg^−1^ and 15 g Kg^−1^ laminarin was supplemented into diets, respectively. Different letters on the bars indicate statistically significant differences (*p* < 0.05).

## Data Availability

All 16s amplicon sequencing data have been submitted to NCBI with the accession number PRJNA904939. Other data not presented in article are available upon reasonable request.

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
