# Peer review of "An Evaluation of Laminarin Additive in the Diets of Juvenile Largemouth Bass (Micropterus salmoides): Growth, Antioxidant Capacity, Immune Response and Intestinal Microbiota"

_animals, 2023, doi:10.3390/ani13030459_

Round 1
Reviewer 1 Report
The present study aimed to investigate the effects of laminarin dietary additive on growth antioxidant properties, immune response and microbiota of the intestine in Micropterus salmoides.
The current manuscript provides some novelty although the experimental period of 28 days is too short for such studies and cannot support solid conclusions. Nevertheless, this study investigates a topic of actual interest on how some new additives can be beneficial for fish growth and health performance.
The authors must review their manuscript carefully and check for English syntax and grammar errors. In general, there are many places where the English grammar is flawed and there are also spelling errors. It is strongly suggested authors employ an English professional editing service Only a few examples are mentioned here i.e., lines 11, 31, 44, 49, 53, 56, 58, 113, 127, 138, 151-152, 236, 238-242, 249, 252, 260-261, 272, 282, 284.
Specific comments
Correct the word ‘trail’ by ‘trial’ and ‘physique by physiology’ all over the manuscript
Line: 16: ‘in the lipid and ash content of fish flesh’
Line 18: ‘antioxidant capacity of fish’
Line 35: Laminarin (LAM) is a storage β-glucan..’ what do you mean by ‘storage β-glucan’?
Line 37: ‘properties’ instead of ‘effects’
Line 42: …lysozyme (LZM)
Line 67: please add if it is possible some indicative/general information about the production of laminarin
Line 96: please describe the storage conditions of the fish tissues for every analysis
Line 111: please describe briefly the method you have used to isolate RNA from your samples
Line 237, add the word ‘fish’ before the Latin names of the mentioned fish
Line 248: ‘functions’ instead of ‘performance’
Table 1: you have added Cr2O3 which is an inert marker in studies of apparent digestibility measurements. However, no such data are presented in your study
Table 1 line 82: ‘calcium pantothenate’ is better than ‘megapantho’
Author Response
Response to Reviewer 1 Comments
Dear reviewer:
Thank you very much for your comments about our paper “An evaluation of laminarin additive in the diets of juvenile largemouth bass (Micropterus salmoides): Growth, antioxidant capacity, immune response and intestinal microbiota”, which were helpful for improving our paper’s quality.
We have considered the comments and made corrections carefully. Here we submit a revised manuscript with trace change as well as the detailed responses. We appreciate for your warm work earnestly, and hope that the revised manuscript will meet with approval. The following is our response to each comment point by point.
Point 1: The authors must review their manuscript carefully and check for English syntax and grammar errors. In general, there are many places where the English grammar is flawed and there are also spelling errors. It is strongly suggested authors employ an English professional editing service Only a few examples are mentioned here i.e., lines 11, 31, 44, 49, 53, 56, 58, 113, 127, 138, 151-152, 236, 238-242, 249, 252, 260-261, 272, 282, 284.
Response: Thanks very much for your suggestion. We have invited a native English expert to help us carefully check and revise the English syntax, grammar and spelling errors all over the manuscript. Please check the “Track change” in the revised manuscript.
Point 2: Correct the word ‘trail’ by ‘trial’ and ‘physique by physiology’ all over the manuscript
Response: Thanks very much for your suggestion. We have revised the word ‘trail’ by ‘trial’ in Line12 and 71; revised the word ’physique’ by ’physiology’ in Line 37, 56 and 107.
Point 3: Line: 16: ‘in the lipid and ash content of fish flesh’
Response: Thanks very much for your suggestion. We have revised this sentence to “as well as in the lipid and ash content of fish flesh” in Line 18.
Point 4: Line 18: ‘antioxidant capacity of fish’
Response: Thanks very much for your suggestion. We have revised this description in Line 20-21.
Point 5: Line 35: Laminarin (LAM) is a storage β-glucan..’ what do you mean by ‘storage β-glucan’?
Response: Sorry, it’s our fault. We have canceled the worf ’storage’ in Line 38.
Point 6: Line 37: ‘properties’ instead of ‘effects’
Response: Thanks very much for your suggestion. We have substitute the word of “effects” by “properties” in Line 40.
Point 7: Line 42: …lysozyme (LZM)
Response: Sorry, it’s our fault. We have supplemented the word of lysozyme in Line 50.
Point 8: Line 67: please add if it is possible some indicative/general information about the production of laminarin
Response: Thanks very much for your suggestion. We have added the purity and source of laminarin in Line 76-78 as follows: “Laminarin is purchased from Xiya Reagent (Xiya Reagent Co., Ltd. Linyi, China) with a purity at leas of 99.5%, and laminarin is extracted from Laminaria digitate”.
Point 9: Line 96: please describe the storage conditions of the fish tissues for every analysis
Response: Sorry, it’s our fault. We have supplemented the storage conditions of the fish tissues for every analysis in Line 106-112, as follows:“Twenty-four fish per replicate were dissected under sterile conditions to pull out the intestine, and then the intstine was cut into small pieces and washed with phosphate buffered saline (PBS) (pH 7.5) to harvest the intstinal tissues and contents. Then, both of them were immediately stored at−80°C in TRIzol reagent (Tiangen, Beijing, China) for RNA extraction. Nine fish per replicate were sampled to frozen by liquid nitrogen, and was stored in −80℃refrigerator for the analysis of flesh composition and antioxidant capacity, respectively.”
Point 10: Line 111: please describe briefly the method you have used to isolate RNA from your samples
Response: Thanks very much for your suggestion. We have described the method of RNA isolation in Line 124-128, as follows: “Total RNA was extracted from the intetinal tissues using Trizol reagent (Tiangen, Beijing, China). The final RNA was eluted in an appropriate amount of 0.1% diethyl pyrocarbonate (DEPC) treated water (SigmaeAldrich, St. Louis, MO, USA). The RNA amount was determined using a Nanodrop 2000. Then the cDNA was synthesized with the reverse transcription”.
Point 11: Line 237, add the word ‘fish’ before the Latin names of the mentioned fish
Response: Thanks very much for your suggestion. We have added the fish before its Latin names in whole manuscript.
Point 12: Line 248: ‘functions’ instead of ‘performance’
Response: Thanks very much for your suggestion. We have substitute the word of “performance” by “functions”. in Line 273.
Point 13: Table 1: you have added Cr2O3 which is an inert marker in studies of apparent digestibility measurements. However, no such data are presented in your study.
Response: Sorry its our fault, we designed this parameter of apparent digestibility before the feeding experiment , while at the end of the experiment, we did not obtain relevant data due to our operational failure.
Point 14: line 82: ‘calcium pantothenate’ is better than ‘megapantho’
Response: Thanks very much for your suggestion. We have substitute the word of “megapantho” by “calcium pantothenate” in Line 92.

Reviewer 2 Report
Laminarin is kind of glucan that found in brown algae and this low-molecular-weight polysaccharide can be a potential prebiotic in aquaculture nutrition. Therefore, this study is innovative and worthy. However, the major problems are the design study in the M&M section and the discussion section. I spend a lot of time to improve the manuscript and the drawbacks are listed here as a follow:
§ L15: how many fish? how many tanks? What was the initial weight?
§ L16: Explain the acronyms fully for the first time. Please check it throughout the manuscript.
§ L17: Plz indicate with antioxidant enzymes were significantly changed
§ L22: …. higher than the control group.
§ L24: …. in largemouth bass.
§ L29: Plz remove LMB
§ L37: How LAM can exert these biological activities in the host? are they related to the chemical functional group? If yes, please provide some information in this context.
§ L45: Do not forget to italic the scientific names
§ L46: …. in the fish diet.
§ L67: It is crucial to clarify for the source of LAM: What was the purity? Form what microalgae species it was extracted? by which method?
§ L69: What is the criteria for choosing the LAM doses and based on the available literature for LAM, why the authors chose the higher doses (more than 5 g/kg) and finally why the chose a high interval among the doses?
§ L92: The duration of the feeding trial (28 days) is lower than a standard period of time to see and conclude on the results. Because it is proven that at least an 8-week period of time is needed to recognize the effects of treated feed on growth performance and physiological aspects in fish species under captivity conditions. What is your scientific justification(s)?
§ Table 1: fishmeal is correct instead of fish meal
§ Table 1: The diets are not balance and sum of the formula was more than 1000 g ! It is not acceptable at all, plz double check
§ L92: How could you be sure to reach the apparent satiation?
§ L96: What was the anesthesia dose? How long it takes to anesthesia the fish?
§ L97: 21 fish per tank ?! Did you pooled the samples? If yes, the authors should clarify through the text. Why the authors selected a large amount of specimens? According to the fish size in this study, I thought analyzing three fish separately from each tank (replicate) was enough. Any justification(s)?
§ L105: are they (diagnostic kits) characterized for human? no problem to use commercial kits. But the authors should explain 1)a short methods and wavelength in each method 2) notice that these kits were previously used for fish
§ Table 2: Plz provide the accession numbers, Tm, and product size.
§ L115: One reference citation is enough for delta Ct analysis.
§ L118: Plz provide the sampling of the intestine first. For instance, from which region (fore, mid, etc)? and how?
§ L128: is there any scientist justification(s) to use Duncan’s multiple range test instead of Tukey or LSD, which are more accurate?
§ L130: The authors should clarify that all the diagrams and analysis done with which software (e.g. for PCA, Venn diagram, etc.).
§ The FCR value was significantly improved in LL and ML compared to HL, but the growth rate did not change significantly. Why? Lower FCR values indicate that a feed is efficiently converted into fish weight gain. Plz provide feed intake value for the experimental groups.
§ Out of all the proinflammatory and anti-proinflammatory cytokines why have the authors chosen il-1β, tnf-α 10, and TGF-β? Which pathway or pathways are you focusing? These need to be fully discussed in the discussion section.
§ L247: It was not significant according to the statics, maybe the authors should change their statement throughout the manuscript.
§ L248: Plz justify the growth performance and nutrient efficacy in a mechanistic views according to LAM.
§ L250-246: In the case of indigenous antioxidant enzymes (antioxidant defense system), in contrary to your results, I found recently in above 2018 that there are some limited published papers that showed these enzymes are decreased by administrating feed additive with antioxidant effect (medicinal plants) in fish. Therefore, the authors should state and challenge the results. This make the discussion section interesting for readers and just presenting the supporting articles is not enough.
§ As the authors announced that LAM has a high antioxidant capacity and can reduce various radicals. Therefore, it is a question that when LAM can be directly used in the body to neutralize free radicals, why LAM at a dose of 5.0 g kg-1 remarkably enhanced the T-AOC, SOD and GSH activities?
L310: The authors should propose the polynomial tests for the candidate parameters to find the optimal level of LAM in the fish diet for the future studies.
Author Response
Response to Reviewer 2 Comments
Dear reviewer:
Thank you very much for your comments about our paper “An evaluation of laminarin additive in the diets of juvenile largemouth bass (Micropterus salmoides): Growth, antioxidant capacity, immune response and intestinal microbiota”, which were helpful for improving our paper’s quality.
We have considered the comments and made corrections carefully. Here we submit a revised manuscript with trace change as well as the detailed responses. We appreciate for your warm work earnestly, and hope that the revised manuscript will meet with approval. The following is our response to each comment point by point.
Point 1: L15: how many fish? how many tanks? What was the initial weight?
Response: Thanks very much for your suggestion. The fish number, tank number and fish initial weight was shown in Line 14-15.
Point 2: L16: Explain the acronyms fully for the first time. Please check it throughout the manuscript.
Response: Thanks very much for your suggestion. We have carefully revised the acronyms all the manuscript. Please check the “Track change” in the revised manuscript.
Point 3: L17: Plz indicate with antioxidant enzymes were significantly changed
Response: Sorry, it’s our fault. We have indicated the significantly changed of antioxidant enzymes in Line 20-21.
Point 4: L22: …. higher than the control group.
Response: Sorry, it’s our fault. We have revised this sentence as follows:” while the content of opportunistic pathogen Plesiomonas, Aeromonas and Brevinema in fish of HL group was substantially higher than the control group” in Line 26.
Point 5: L24: …. in largemouth bass.
Response: Sorry, it’s our fault. We have substitute the word of “of” by “in” in Line 27.
Point 6: L29: Plz remove LMB
Response: Thanks very much for your suggestion. We have removed the word ”LMB” in Line 33.
Point 7: L37: How LAM can exert these biological activities in the host? are they related to the chemical functional group? If yes, please provide some information in this context.
Response: Thanks very much for your suggestion. We have supplemented these information in Line 39-47, as follows:”Lines of evidence demonstrated that specific physicochemical properties play a vital role in determining the magnitude of β-glucan binding to macrophage receptor(s) and how it modulates the immune responses. Moreover, published reports has demonstrated that LAM is featured by antioxidant, immunopotentiator, antitumor and antivirus properties. Moreover, many research revealed that LAM displayed immune-modulatory effects in fish because of its binding capacity to different receptors on leukocytes leading to the stimulation of immune responses including bactericidal activity, cytokine productivity, and survival fit ability at cellular levels”.
Point 8: L45: Do not forget to italic the scientific names
Response: Sorry, it’s our fault. We have checked the italic the scientific names all through the manuscript, Please check the “Track change” in the revised manuscript.
Point 9: L46: …. in the fish diet.
Response: Sorry, it’s our fault. We have added “the” in this sentence in Line 50.
Point 10: L67: It is crucial to clarify for the source of LAM: What was the purity? Form what microalgae species it was extracted? by which method?
Response: Thanks very much for your suggestion. We have supplemented these information in Line 72-73 as follows: Laminarin is purchased from Xiya Reagent (Xiya Reagent Co., Ltd. Linyi, China) with a purity at leas of 99.5%, and laminarin is extracted from Laminaria digitate using the warm-water extraction method.
Point 11: L69: What is the criteria for choosing the LAM doses and based on the available literature for LAM, why the authors chose the higher doses (more than 5 g/kg) and finally why the chose a high interval among the doses?
Response: Thanks very much for your suggestion. Previous report has demonstrated that dietary high dose of laminarin inclusion (8 g/kg laminarin in basal diet ) significantly increased the tail length, body weight and immune related gene expression, indicating that the high dose of laminarin (8 g in dietary) changed fish to be stronger (Aquacul Nutr 2021, 27, 1181-1191). Addition of 6 g/kg laminarin in basal diet would improve the growth performance and immunity of pearl gentian groupers in aquaculture (Anim Husb Feed Sci 2017, 9, 259-262). So, we chose the the higher doses (more than 5 g/kg) is workable. Moreover, previous literature chose the dose (basal diet supplemented with 0.5%, 1.0%, and 1.5% laminarin) in groupers (Fish & Shellfish Immunology 41 (2014) 402-406), therefore, we think the high interval among the doses in this study is also reasonable.
Point 12: L92: The duration of the feeding trial (28 days) is lower than a standard period of time to see and conclude on the results. Because it is proven that at least an 8-week period of time is needed to recognize the effects of treated feed on growth performance and physiological aspects in fish species under captivity conditions. What is your scientific justification(s)?
Response: Thanks very much for your suggestion. In this field, 56-day feeding trial is a standard period of time to conclude on the results, while, when some of previous reports has also demonstrated that fed fish with laminarin-supplemented diet (0.2 g/kg/day) produced an increase in the phagocytic activity in the head kidney macrophages and a significant increase in the secretion of TNFa and IL-8 of rainbow trout at 21 days (Fish Shellfish Immunology 34 (2013) 1692-1752). Moreover, dietary β-glucan significantly increased phagocytosis, superoxide anion production and SOD activity of sea cucumbers (P < 0.05) at 28 days (Fish Shellfish Immunology 31 (2011) 303-309). In this study, we focused on the effect of laminarin as an immunopotentiator, therefore, a 28-day feeding trail is enough to investigate the effect of laminarin as on fish immunity.
Point 13: Table 1: fishmeal is correct instead of fish meal
Response: Thanks very much for your suggestion. We have substitute “fish meal” by “fishmeal” in Table 1
Point 14: Table 1: The diets are not balance and sum of the formula was more than 1000 g ! It is not acceptable at all, plz double check
Response: Thanks very much for your suggestion. In this experiment, laminarin was used as an additive in basal diet, but not substitute other component. The similar supplementation was also be described in previous literature, such as “Aquacul Nutr 2021, 27, 1181-1191”, “Fish Shellfish Immun, 2014, 41, 402-406”.
Point 15: L92: How could you be sure to reach the apparent satiation?
Response: Thanks very much for your suggestion. As a fierce carnivorous fish, juvenile largemouth bass stop predating when reach satiation.
Point 16: L96: What was the anesthesia dose? How long it takes to anesthesia the fish?
Response: Thanks very much for your suggestion.The dose of MS-222 used in this experiment is 55 mg/L. It takes 250s-300s to anesthesia the fish.
Point 17: L97: 21 fish per tank ?! Did you pooled the samples? If yes, the authors should clarify through the text. Why the authors selected a large amount of specimens? According to the fish size in this study, I thought analyzing three fish separately from each tank (replicate) was enough. Any justification(s)?
Response: Thanks very much for your suggestion. We have stated that 480 juvenile LMB were randomly assigned to four groups (40 fish per tank with 3 replicates in each group) in Line 96-97. The reason why we selected a large amount of specimens to analyze is due to the single intestine is small and the intestinal content is little for microbiome analysis.
Point 18: L105: are they (diagnostic kits) characterized for human? no problem to use commercial kits. But the authors should explain 1)a short methods and wavelength in each method 2) notice that these kits were previously used for fish
Response: Thanks very much for your suggestion. These kits are characterized for fish. Large number of literature have reported that these commercial kit very mature and effective, which have been widely used in fish research, such as “Aquaculture 534 (2021) 736261”, “Fish and Shellfish Immunology 120 (2022) 706–715”, “Aquaculture Reports 21 (2021) 100823”, “Front. Immunol. 12:827946”, “Aquaculture Reports 22 (2022) 100954”, etc.
Point 19: Table 2: Plz provide the accession numbers, Tm, and product size.
Response: Thanks very much for your suggestion. We have supplemented the accession numbers, Tm, and product size in Table 2.Please check the Table 2, thanks.
Point 20: L115: One reference citation is enough for delta Ct analysis.
Response: Thanks very much for your suggestion. We have canceled two references for delta Ct analysis in Line 126.
Point 21: L118: Plz provide the sampling of the intestine first. For instance, from which region (fore, mid, etc)? and how?
Response: Thanks very much for your suggestion. We have supplemented the Twenty-four fish per replicate were dissected under sterile conditions to pull out the intestine, and then the mid intestine was cut into small pieces and washed with phosphate buffered saline (PBS) (pH 7.5) to harvest the intstinal tissues and contents. Then, both of them were immediately stored at−80°C in TRIzol reagent (Tiangen, Beijing, China) for RNA extraction in Line 103-108.
Point 22: L128: is there any scientist justification(s) to use Duncan’s multiple range test instead of Tukey or LSD, which are more accurate?
Response: Thanks very much for your suggestion. Duncan’s multiple range test is a modification of SNK method, but it improves the probability of Class I errors and reduces the probability of Class II errors, which is widely used in agricultural research including fish.
Point 23: L130: The authors should clarify that all the diagrams and analysis done with which software (e.g. for PCA, Venn diagram, etc.).
Response: Thanks very much for your suggestion.The intestinal microbiota analysis including Principal component analysis (PCA), Venn diagram, Microbial community bar plots (MCBP) and Linear discriminant analysis Effect Size (LEfSe) were performed using the BMKCloud software (www.biocloud.net) in Line 134-138.
Point 24: The FCR value was significantly improved in LL and ML compared to HL, but the growth rate did not change significantly. Why? Lower FCR values indicate that a feed is efficiently converted into fish weight gain. Plz provide feed intake value for the experimental groups.
Response: Thanks very much for your suggestion. We have supplemented the total feed intake in Table 3. The SGR did not change significantly, while the FCR value was significantly decreased in LL and ML compared to HL, indicating that dietary LAM might enhance the feed digestion and nutrient utilization in LL and ML groups.
Point 25: Out of all the proinflammatory and anti-proinflammatory cytokines why have the authors chosen il-1β, tnf-α 10, and TGF-β? Which pathway or pathways are you focusing? These need to be fully discussed in the discussion section.
Response: Thanks very much for your suggestion. Cytokines, including proinflammatory and anti-inflammatory cytokines, are mainly responsible for the host innate defense in fish. The proinflammatory cytokine tnf-a is an important mediator in the regulation of inflammatory response, and its activation induces gene expression of some pro inflammatory factors, such as il-1b in rainbow trout. The anti-inflammatory cytokines, such as tgf-b, can depress the production of pro-inflammatory cytokine, and thereby inhibit the inflammatory response in teleost. Thus, we chose there cytokines as research targets. Receptor-bound β-glucan may mediate the production of inflammatory cytokines (interleukins, interferons and tumor necrosis factor). These signaling proteins are believed to aggravate phagocytic activity of immune cells through oxidative burst and natural cytotoxic liquidation. However, more than 3,000 papers have demonstrated that the effectiveness of β-glucan in improving fish immunity, but the detailed knowledge of the receptors involved in recognizing β-glucans and their downstream signaling mechanism is yet to be clarified in teleosts until now“Molecules 25(2020), 5378”.
Point 26: L247: It was not significant according to the statics, maybe the authors should change their statement throughout the manuscript.
Response: Thanks very much for your suggestion. We have changed the statement of these description throughout the manuscript
Point 27: L248: Plz justify the growth performance and nutrient efficacy in a mechanistic views according to LAM.
Response: Thanks very much for your suggestion. We have not understood your concern about ”justify the growth performance and nutrient efficacy in a mechanistic views mechanistic views”. Moreover, We also have not found this in previous research of fish additive. We wish you can provide more information about this, which can help us improve the manuscript further.
Point 28: L250-246: In the case of indigenous antioxidant enzymes (antioxidant defense system), in contrary to your results, I found recently in above 2018 that there are some limited published papers that showed these enzymes are decreased by administrating feed additive with antioxidant effect (medicinal plants) in fish. Therefore, the authors should state and challenge the results. This make the discussion section interesting for readers and just presenting the supporting articles is not enough.
Response: Thanks very much for your suggestion. In this study, the result showed a tendency of low concentration promoted antioxidant capacity and high concentration inhibition antioxidant capacity. However, reason why higher level of LAM supplementation level in diet decreased the CAT and SOD activity might be due to the feed composition greatly changed, which should be investigated in further research.
Point 29: As the authors announced that LAM has a high antioxidant capacity and can reduce various radicals. Therefore, it is a question that when LAM can be directly used in the body to neutralize free radicals, why LAM at a dose of 5.0 g kg-1 remarkably enhanced the T-AOC, SOD and GSH activities?
Response: Thanks very much for your suggestion. The reason why dietary LAM at a dose of 5.0 g kg-1 remarkably enhanced the T-AOC, SOD and GSH activities might be due to LAM boost up the hosts non-specific defense mechanism and instigate leukocytes triggering their phagocytic and anti-pathogenic reactions through production of pro-active oxygen species (superoxide anions, hydrogen peroxide, hydroxyl radical, hypochlorous acid, etc.) and nitrogen intermediates (nitrite, amides, and nitrogen dioxide).
Point 30: The authors should propose the polynomial tests for the candidate parameters to find the optimal level of LAM in the fish diet for the future studies.
Response: Thanks very much for your suggestion. We had considered the polynomial tests for the candidate parameters. While, in this study, on account of the measurement of multiple parameters including growth, antioxidant capacity, immune and microbiota, etc., we think one-way ANOVA test is more workable.

Round 2
Reviewer 2 Report
The authors have made a good effort to improve their manuscript and all comments are addressed sufficiently. Please consider the minor comments as a follow:
According to the previous comment, plz indicate in the M&M section that the intestine samples of how many fish were pooled for each replication (tank).
It is still strange for me that the FCR value was improved and the authors provide a justification that “dietary LAM might enhance the feed digestion and nutrient utilization in LL and ML groups”. Therefore, efficient nutrient utilization is linked to better muscle growth rate. However, the growth rate and weight gain did not significantly changed. In my opinion, the somatic growth could not be manifested significantly due to the short period of feeding trial in this project, maybe if the authors continued the feeding trial for example for 8 weeks, the growth indices would be changed. If you agree with my opinion, please mention this issue in the discussion.
Moreover, plz provide justification for CF (condition factor), because this index was changed significantly among the treatments.
Plz indicate the Tm in Table 2 without decimals.
According to the previous comment, for your help, it is recommended to review the discussion section of the following paper (https://doi.org/10.1016/j.aquaculture.2021.737296) to support the statement of “a tendency of low concentration promoted antioxidant capacity and high concentration inhibition antioxidant capacity” and “The reason why dietary LAM at a dose of 5.0 g/kg remarkably enhanced the T-AOC, SOD and GSH activities”. You have to link the possible expression levels of the relevance antioxidant genes with the levels of the serum remained antioxidant enzymes along with the antioxidant properties of the studied bioactive compounds from your herb (macroalgae).
According to the previous comment, at the end of the conclusion section, since the authors reach to a decision that LL (5 g/Kg) is the best dose, for future studies they can propose to use lower concentration in a dose study manner to find the optimum dietary level of laminarin in largemouth bass.
Author Response
Response to Reviewer 2 Comments (Round 2)
Dear reviewer:
Thank you very much for your comments (Round 2) about our paper “An evaluation of laminarin additive in the diets of juvenile largemouth bass (Micropterus salmoides): Growth, antioxidant capacity, immune response and intestinal microbiota”, which were helpful for improving our paper’s quality.
We have considered the comments and made corrections carefully. Here we submit a revised manuscript with trace change as well as the detailed responses. We appreciate for your warm work earnestly, and hope that the revised manuscript will meet with approval. The following is our response to each comment point by point.
Point 1: According to the previous comment, plz indicate in the M&M section that the intestine samples of how many fish were pooled for each replication (tank).
Response: Thanks very much for your suggestion. Twenty-four fish per replicate (tank) were dissected under sterile conditions to pull out the intestine. Four samples (each sample contains 6 fish) in each replicate (tank) were selected for 16sRNA sequence analysis in Line 117-121.
Point 2: It is still strange for me that the FCR value was improved and the authors provide a justification that “dietary LAM might enhance the feed digestion and nutrient utilization in LL and ML groups”. Therefore, efficient nutrient utilization is linked to better muscle growth rate. However, the growth rate and weight gain did not significantly changed. In my opinion, the somatic growth could not be manifested significantly due to the short period of feeding trial in this project, maybe if the authors continued the feeding trial for example for 8 weeks, the growth indices would be changed. If you agree with my opinion, please mention this issue in the discussion.
Response: Thanks very much for your suggestion. We have supplemented this issue “these discrepancies may be attributable to the fish species difference, as well as the short period of feeding trial in this study, where the effect of LAM on juvenile largemouth bass in a longer feeding trail (8 weeks or more) should be investigated in further research.”in the “discussion” in Line 280-284
Point 3: Moreover, plz provide justification for CF (condition factor), because this index was changed significantly among the treatments.
Response: Thanks very much for your suggestion. In this study, CF (condition factor) was significantly changed between fish in LL and HL group, LL and ML group respectively (Fish in LL group exhibited considerably higher condition factor (CF) than that of fish in ML and HL groups (p < 0.05). “In addition, juvenile fish in LL group exhibited significantly higher CF than that of fish in ML and HL groups (p < 0.05), which might be due to the higher concentration of crude protein and lipids in diet of LL group.” in Line 285-288.
Point 4:According to the previous comment, for your help, it is recommended to review the discussion section of the following paper (https://doi.org/10.1016/j.aquaculture.2021.737296) to support the statement of “a tendency of low concentration promoted antioxidant capacity and high concentration inhibition antioxidant capacity” and “The reason why dietary LAM at a dose of 5.0 g/kg remarkably enhanced the T-AOC, SOD and GSH activities”. You have to link the possible expression levels of the relevance antioxidant genes with the levels of the serum remained antioxidant enzymes along with the antioxidant properties of the studied bioactive compounds from your herb (macroalgae).
Response: Thanks very much for your suggestion. We will follow your advice to investigate the relevance between antioxidant genes, serum remained antioxidant enzymes and antioxidant properties laminarin in further research in further research. The mechanism of how antioxidant properties of LAM affect the fish antioxidant genes expression and antioxidant capacity is another massive systematic work which needs a lot of time to study.
Point 5: According to the previous comment, at the end of the conclusion section, since the authors reach to a decision that LL (5 g/Kg) is the best dose, for future studies they can propose to use lower concentration in a dose study manner to find the optimum dietary level of laminarin in largemouth bass.
Response to Reviewer 2 Comments (Round 2)
Dear reviewer:
Thank you very much for your comments (Round 2) about our paper “An evaluation of laminarin additive in the diets of juvenile largemouth bass (Micropterus salmoides): Growth, antioxidant capacity, immune response and intestinal microbiota”, which were helpful for improving our paper’s quality.
We have considered the comments and made corrections carefully. Here we submit a revised manuscript with trace change as well as the detailed responses. We appreciate for your warm work earnestly, and hope that the revised manuscript will meet with approval. The following is our response to each comment point by point.
Point 1: According to the previous comment, plz indicate in the M&M section that the intestine samples of how many fish were pooled for each replication (tank).
Response: Thanks very much for your suggestion. Twenty-four fish per replicate (tank) were dissected under sterile conditions to pull out the intestine. Four samples (each sample contains 6 fish) in each replicate (tank) were selected for 16sRNA sequence analysis in Line 117-121.
Point 2: It is still strange for me that the FCR value was improved and the authors provide a justification that “dietary LAM might enhance the feed digestion and nutrient utilization in LL and ML groups”. Therefore, efficient nutrient utilization is linked to better muscle growth rate. However, the growth rate and weight gain did not significantly changed. In my opinion, the somatic growth could not be manifested significantly due to the short period of feeding trial in this project, maybe if the authors continued the feeding trial for example for 8 weeks, the growth indices would be changed. If you agree with my opinion, please mention this issue in the discussion.
Response: Thanks very much for your suggestion. We have supplemented this issue “these discrepancies may be attributable to the fish species difference, as well as the short period of feeding trial in this study, where the effect of LAM on juvenile largemouth bass in a longer feeding trail (8 weeks or more) should be investigated in further research.”in the “discussion” in Line 280-284
Point 3: Moreover, plz provide justification for CF (condition factor), because this index was changed significantly among the treatments.
Response: Thanks very much for your suggestion. In this study, CF (condition factor) was significantly changed between fish in LL and HL group, LL and ML group respectively (Fish in LL group exhibited considerably higher condition factor (CF) than that of fish in ML and HL groups (p < 0.05). “In addition, juvenile fish in LL group exhibited significantly higher CF than that of fish in ML and HL groups (p < 0.05), which might be due to the higher concentration of crude protein and lipids in diet of LL group.” in Line 285-288.
Point 4: Plz indicate the Tm in Table 2 without decimals.
Response: Thanks very much for your suggestion. Its our fault, we have revised the Tm without decimals in Table 2.
Point 5:According to the previous comment, for your help, it is recommended to review the discussion section of the following paper (https://doi.org/10.1016/j.aquaculture.2021.737296) to support the statement of “a tendency of low concentration promoted antioxidant capacity and high concentration inhibition antioxidant capacity” and “The reason why dietary LAM at a dose of 5.0 g/kg remarkably enhanced the T-AOC, SOD and GSH activities”. You have to link the possible expression levels of the relevance antioxidant genes with the levels of the serum remained antioxidant enzymes along with the antioxidant properties of the studied bioactive compounds from your herb (macroalgae).
Response: Thanks very much for your suggestion. We will follow your advice to investigate the relevance between antioxidant genes, serum remained antioxidant enzymes and antioxidant properties laminarin in further research in further research. The mechanism of how antioxidant properties of LAM affect the fish antioxidant genes expression and antioxidant capacity is another massive systematic work which needs a lot of time to study.
Point 6: According to the previous comment, at the end of the conclusion section, since the authors reach to a decision that LL (5 g/Kg) is the best dose, for future studies they can propose to use lower concentration in a dose study manner to find the optimum dietary level of laminarin in largemouth bass.
Response: Thanks very much for your suggestion. Although in this study, we just determined the appropriate dose of dietary laminarin is 5 g/Kg for juvenile largemouth bass, but not the optimal dosage. Moreover, we found higher dose of dietary laminarin (15 g/Kg) led to a increase of pathogenic bacteria in of juvenile LMB intestine and further negatively affect fish physiological performance. Overall, Sharing this result will help us or other researchers save a lot of time to select more accurate dose range in further study.
